# High-Molecular-Weight Dextran-Type Exopolysaccharide Produced by the Novel *Apilactobacillus waqarii* Improves Metabolic Syndrome: In Vitro and In Vivo Analyses

**DOI:** 10.3390/ijms232012692

**Published:** 2022-10-21

**Authors:** Waqar Ahmad, Jacqueline L. Boyajian, Ahmed Abosalha, Anam Nasir, Iram Ashfaq, Paromita Islam, Sabrina Schaly, Rahul Thareja, Azam Hayat, Mujaddad ur Rehman, Munir Ahmad Anwar, Satya Prakash

**Affiliations:** 1Biomedical Technology and Cell Therapy Research Laboratory, Department of Biomedical Engineering, Faculty of Medicine, McGill University, 3775 University Street, Montreal, QC H3A 2B4, Canada; 2Industrial Biotechnology Division, National Institute for Biotechnology and Genetic Engineering College, Pakistan Institute of Engineering and Applied Sciences (NIBGE-C, PIEAS), Faisalabad 38000, Pakistan; 3Department of Microbiology, Abbottabad University of Science and Technology, Havelian, Abbottabad 22500, Pakistan; 4Pharmaceutical Technology Department, Faculty of Pharmacy, Tanta University, Tanta 31111, Egypt

**Keywords:** *Apilactobacillus waqarii*, prebiotics, dextran, intestinal health, metabolic syndrome, body weight, blood glucose, serum cholesterol

## Abstract

Metabolic syndrome is a leading medical concern that affects one billion people worldwide. Metabolic syndrome is defined by a clustering of risk factors that predispose an individual to cardiovascular disease, diabetes and stroke. In recent years, the apparent role of the gut microbiota in metabolic syndrome has drawn attention to microbiome-engineered therapeutics. Specifically, lactic acid bacteria (LAB) harbors beneficial metabolic characteristics, including the production of exopolysaccharides and other microbial byproducts. We recently isolated a novel fructophilic lactic acid bacterium (FLAB), *Apil**actobacillus waqarii* strain HBW1, from honeybee gut and found it produces a dextran-type exopolysaccharide (EPS). The objective of this study was to explore the therapeutic potential of the new dextran in relation to metabolic syndrome. Findings revealed the dextran’s ability to improve the viability of damaged HT-29 intestinal epithelial cells and exhibit antioxidant properties. In vivo analyses demonstrated reductions in body weight gain and serum cholesterol levels in mice supplemented with the dextran, compared to control (5% and 17.2%, respectively). Additionally, blood glucose levels decreased by 16.26% following dextran supplementation, while increasing by 15.2% in non-treated mice. Overall, this study displays biotherapeutic potential of a novel EPS to improve metabolic syndrome and its individual components, warranting further investigation.

## 1. Introduction

Metabolic syndrome is a non-communicable disease that is spreading rapidly across the globe. According to the World Health Organization (WHO), metabolic syndrome is defined as the presence of insulin resistance in combination with two or more metabolic risk factors, including obesity, hyperlipidemia, hypertension or microalbuminuria [1]. Individuals with metabolic syndrome have an increased risk of cardiovascular disease, type 2 diabetes, stroke and all-cause mortality, as well as health-related poor quality of life [2,3,4]. The underlying etiology of metabolic syndrome is complex yet mainly governed by a sedentary lifestyle, poor diet and obesity [5]. The prevalence of metabolic syndrome often corresponds with that of obesity, making the rising obesity pandemic a great concern.

Current treatment strategies for metabolic syndrome include lifestyle and dietary interventions, surgery and pharmacological interventions [6,7,8]. However, the chronic nature of metabolic syndrome requires long-term management of the various risk factors. In addition, the lack of a single drug to treat metabolic syndrome necessitates a polypharmacological approach to manage its individual components [8]. The long-term use and combination of multiple drugs to manage metabolic syndrome often leads to deleterious side effects, poor patient compliance and drug–drug interactions [9]. Thus, the use of natural compounds to treat and/or manage metabolic syndrome is of growing interest.

There is recent discussion on the role of bioactive compounds in the management of metabolic syndrome [10]. Notably, prebiotics, probiotics and synbiotics are among the most important bioactive ingredients in the functional food market [11]. Prebiotics are substrates that are selectively utilized by host microorganisms conferring a health benefit [12]. Probiotics are live microorganisms that, when administered in adequate amounts, confer a health benefit to the host [13]. Prebiotics can be used with microbes to efficiently utilize the provided energy source. This combination, known as synbiotics, provides synergistic effects to the host [14]. Recently, prebiotics such as levans, inulin, oligosaccharides, resistant starch, dextran, pyrodextrins, and lactosucrose are often added to functional foods to enhance the activity of the nutritional ingredients, acting as innovative and important health promoters of bioactive food products [15,16]. Prebiotics have the potential to improve human and animal health by controlling the imbalance of the gut microbiota through the stimulation of the growth of beneficial bacteria. Specifically, *Lactobacillus* and *Bifidobacterium* inhibit the proliferation of harmful bacteria and increase the production of short-chain fatty acids, such as acetate, butyrate and propionate, when there is an increase in healthy bacteria [12,17,18]. In turn, this results in the improvement of intestinal membrane integrity and absorption of nutrients, lowered glycemic levels and body weight, decreased cholesterol, improved immunity and modulation of metabolic and cardiovascular parameters [19,20,21].

The gut microbiota plays a critical role in the development and progression of metabolic syndrome and may therefore be a target for the treatment of metabolic syndrome [22]. An imbalance of the ratios between the main phyla, known as microbial dysbiosis, is a hallmark of several diseases, including metabolic syndrome and its individual components [22]. Dietary intake of probiotics, such as lactic acid bacteria (LAB), may modulate the gut microbiota and help improve the dysbiotic state, thereby promoting metabolic homeostasis [23]. Additionally, the beneficial effects of prebiotics on metabolic syndrome are demonstrated in previous pre-clinical and human studies [24]. Oligofructose, inulin and xylo-oligosaccharides are among other prebiotics that attenuate key markers of obesity through restoring the presence of beneficial gut bacteria (e.g., *Bifidobacterium*), improving gut barrier integrity and therefore reducing metabolic endotoxemia and systemic inflammation. Short-chain fatty acid (SCFA) fermentation from prebiotic fibers may also have anti-obesogenic effects, such as bacterial-derived propionate. Overall, the use of beneficial probiotics such as LAB, prebiotic fibers and microbial byproducts are proving to have a strong influence on the treatment and/or management of metabolic syndrome. Thus, the investigation of novel bacterial strains can promote the development of microbiome-engineered therapeutics for metabolic disorders.

Towards this end, we previously isolated a novel fructophilic lactic acid bacteria (FLAB) *Apilactobacillus waqarii* (HBW1) strain from giant honeybee gut and identified its production of a dextran-type exopolysaccharide (EPS). Dextran is a homopolysaccharide composed of D-glucose units containing α-(1,6) linkages in the main chain and varied percentages of α-(1,2), α-(1,3), α-(1,4) branch linkages [25]. These branched linkages vary from strain to strain. For example, the dextran produced by *Leuconostic mesenteroides* FT045 B isolated from an alcohol and sugar mill plant contains 97.9% α-(1,6) linkages and 2.1% α-(1,3) branch linkages [26]. Moreover, *Leuconostic citreum* NM105 synthesizes dextran and contains 32% α-(1,2) branching linkages [27]. Two exopolysaccharides (EPS) produced by the native *Leuconostic pseudomesenteroides* strain were identified as linear which only contained α-(1,6) linkages without additional branching [28]. Considering the positive effects of probiotics and prebiotics on metabolic regulation, we sought to explore the beneficial properties of the novel dextran for the treatment of metabolic syndrome risk factors.

In the present study, EPS produced by the newly isolated HBW1 strain is characterized and investigated for therapeutic benefits in metabolic syndrome. The dextran-type EPS can reduce weight gain, serum glucose and cholesterol levels, acting as a promising biotherapeutic for the improvement of metabolic syndrome and its individual components.

## 2. Results and Discussion

### 2.1. Optimization of Process Parameters for Exopolysaccharide Production by Bacteria

The isolate HBW1 produced a maximum dextran yield of 24 g/L after optimizing different culturing conditions. The maximum yield was obtained by shaking incubation (150 rpm) at 30 °C, pH 7 and substrate concentration of 250 g/L, as seen in Figure 1. This isolate produced a high amount of EPS as compared to previously reported values from other species, including *Fructobacillus fructosus* N10 which produces 4.5 g/L [29]; *Leuconostoc mesenteroides* N5 with 9.8 g/L [29]; *Leuconostoc pseudomesenteroides* yielding 12.5 g/L [30]; and *Leuconostoc ctireum* NM105 making 23.5 g/L [27].

### 2.2. Monosaccharide Composition of the Exopolysaccharide

With regard to retention time, a major peak at 8.753 min represented glucose after comparing with the standard (sulfuric acid RT 15.130), as shown in Figure 2. Based on the peak area, EPS from HBW1 was determined to be composed of glucose as a main component, with a total glucan content of approximately 91.73%. The total sugar content in HBW1 EPS is found to be higher than previously reported LAB such as *Fructobacillus fructosus* N10 (70%) and *Fructobacillus fructosus* N4 (61%) [29]. The moisture level in the dried EPS was noted as 16.50% of the total EPS content.

### 2.3. ^13^C NMR Spectroscopy of the Bacterial-Derived Glucan

The ^13^CNMR spectroscopic signals provide convincing results, corresponding to the important features particular for dextran-type glucan. The chemical shift values in the spectra for the glucan synthesized by HBW1 shows six resonance signals at 100.43 (C1), 74.12 (C2), 76.11 (C3), 72.90 (C4), 72.24 (C5) and 68.25 (C6) ppm (Figure 3). These chemical shift values are in agreement with the peak position for dextran from *Leuconostoc mesenteroides* CMG713 [31], *Leuconostoc pseudomesenteroides* R2 [28] and *Weissella cibaria* CMGDEX3 [32], representing similar structure.

### 2.4. Apilactobacillus waqarii Dextran-Type Glucan Linkage Analysis

The identity of each carbohydrate derivative was determined by using criteria previously discussed [33], as well as by comparison with the gas chromatography (GC) with electron impact mass spectrometry (GC-EIMS) of partially methylated alditol acetates (PMAAs) database offered by Complex Carbohydrate Research Center [34]. Figure 4 shows the GC chromatogram of the PMAAs of HBW1 polysaccharide. A signal peak at 8.657 min was identified as t-Glcp: 1,5-Di-O-acetyl-2,3,4,6-tetra-O-methyl-D-glucitol. The signal peak at 10.635 min confirmed the presence of α-(1→6) linked Glcp residues which yielded 1,5,6-Tri-O-acetyl-2,3,4-tri-O-methyl-D-glucitol sugar derivative. Additionally, a signal peak at 12.607 min confirmed the presence of 1,3,5,6-Tetra-O-acetyl-2,4-di-O-methyl-D-glucitol, representing α(1→3,6) linked Glcp residues. The peak with the highest retention time at 18.342 min was designated to the internal myo-inositol standard. The molar ratio of each sugar residue was calculated from the peak area in the GC chromatogram, as demonstrated in Table 1. Visibly, (1→6) linked Glcp was the most abundant linkage type, which accounts for 66.74% of the residues. In addition, one PMAA representing a major branching of the polysaccharide backbone at position O3 (1→3,6 linked Glcp) with molar ratio of 11.76% was detected. Another non-reducing terminal unit was t-Glcp with the molar ratio of 21.5%. Previously, dextran extracted from deteriorated sugarcane contained 94.3% α-(1→6) linkages and 1.31% α-(1→3,6) branched linkages [33]. Similar results were observed for dextran from *Leuconostoc mesenteroides* which contained 95% α-(1→6) linkages and 5% α-(1→3,6) branched linkages [35]. Our results, therefore, indicated the presence of a highly branched dextran-type glucan from *Apilactobacillus waqarii*.

### 2.5. Exopolysaccharide Molecular Weight Distribution

The average molecular weight of the purified EPS from HBW1 was 5.46 × 10^8^ Da, which was higher than dextran produced by *Leuconostoc mesenteroides* BD1710 (6.35 × 10^5^ Da) [36], *Leuconostoc pseudomesenteroides* (7.67 × 10^5^ Da) [37] and *Leuconostoc citreum* (6.07 × 10^6^) [38]. Molecular weight of EPS’s is important for studying their structure–function relationship [39]. High molecular weights of EPS’s have been considered essential for promoting the formation of EPS-protein network structure and improving the texture of fermented products, which can be utilized in the industry [28].

### 2.6. Cell Lines Experiments

#### 2.6.1. Intestinal Cell Viability in Response to Exopolysaccharide Exposure

The investigated EPS is non-toxic to intestinal epithelial cells and increased cell viability of HT-29 intestinal epithelial cells. EPS significantly increased cell viability for all treatments, with the highest concentration (5 mg/mL) resulting in the greatest relative increase (Figure 5). Findings show the EPS promotes intestinal cell growth and has no cytotoxic effect, suggesting it to be a potentially safe medicinal agent with therapeutic outcomes for intestinal damage. Previous research shows similar effects of bacterial-derived EPS. Notably, EPS from *Bifidobacterium animalis* improved cell growth in enteropathogenic Escherichia coli-damaged intestinal cells [40]. EPS produced by *Streptococcus thermophilus* and *Lactobacillus plantarum* also protect against intestinal epithelial monolayer disruption in colitis models [41,42]. Intestinal epithelial barrier dysfunction and delayed mucosal wound healing is associated with several inflammatory diseases, including inflammatory bowel diseases and metabolic disorders [43,44]. Impaired intestinal barrier integrity enables the translocation of bacterial endotoxins, such as lipopolysaccharides, into the bloodstream and causes metabolic endotoxemia [23]. Epithelial proliferation is essential for effective wound healing after intestinal injury. However, uncontrolled proliferation of epithelial cells can lead to tumorigenesis [45]. The close relationship between normal wound healing and a stimulated tumor microenvironment is linked by shared inflammatory processes [46]. As the HT-29 cell line originates from human colon adenocarcinoma, promotion of cellular proliferation may inhibit apoptotic activity and support tumorigenesis. In fact, previous research concluded Lactobacilli-derived EPS to be potential anti-cancer agents, following anti-proliferative activity on HT-29 cells [47,48]. In the present study, the EPS increased HT-29 cell viability in a dose-dependent manner. Thus, careful investigation of the mechanisms underlying the EPS-induced cell proliferation is needed to determine its impact on tumor cell growth and apoptosis. Such research can help determine the clinical significance of the EPS. The presented findings therefore show therapeutic promise of the novel EPS in relation to mucosal repair.

#### 2.6.2. Antioxidant Potential of the Exopolysaccharide Produced by *Apilactobacillus waqarii*

The present EPS reduces oxidative stress in damaged intestinal epithelial cells. Findings revealed a decrease in ROS expression in HT-29 cells treated with the EPS, compared to non-treated cells, albeit not significant, as shown in Figure 6. An imbalance of ROS production results in intestinal oxidative stress and contributes to various gastrointestinal (GI) pathologies, including IBD, gastroduodenal ulcers and GI malignancies [49,50]. Oxidative stress and mitochondrial dysfunction from ROS overproduction is also highly implicated in metabolic syndrome [51]. EPS is well-established as an antioxidant compound, however, specific structural features of EPS can influence their antioxidant properties [52]. Our results confirm the antioxidant capacity of the novel EPS, which is aligned with available in vitro and in vivo data on other types of microbial EPS [53,54,55]. Moreover, previous research has suggested that antioxidant properties of EPS behave in a dose-dependent manner, as demonstrated by EPS from wild type and mutant *Weissella confusa* [56]. Antioxidant nutrients may protect against oxidative stress by promoting nitric oxide (NO) synthase activity and subsequently downregulating the superoxide, nicotinamide adenine dinucleotide phosphate oxidase (NADPH) oxidase [57]. Synthesis of NO is necessary for proper metabolic homeostasis and can reduce markers of metabolic syndrome, including high blood pressure [58]. Thus, the identified EPS may help manage metabolic syndrome through oxidant scavenging. Reduction in ROS activity of HT-29 cells may also be indicative of anti-cancer effects of the EPS. Elevated ROS supports a tumor microenvironment and plays a role in the onset, progression and worsening of tumors [59]; the use of antioxidants as tumor suppressors is well-established [60]. However, high levels of ROS in cancer cells may act therapeutically by promoting cell death and antitumorigenic signaling [61]. Various natural compounds have been proposed as potential biotherapeutics for colon cancer following induction of ROS and subsequent apoptosis of HT-29 cells [62,63,64]. Considering the double-sided role of ROS in colon adenocarcinoma, future studies are warranted to determine if the novel EPS can restore the altered redox environment in cancer cells for a beneficial effect.

### 2.7. Investigation of Metabolic Syndrome Pre-Clinical Efficacy in Experimental Animals

#### 2.7.1. Determination of the Exopolysaccharide’s Impact on Body Weight

In order to investigate the effect of the EPS on metabolic markers, mice were treated with the dextran through dietary supplementation for three weeks and compared against non-treated control mice. At the end of experiment (day 21), the increase in average body weight in the control group was 3.6 g compared to 1.7 g in the dextran-fed group (Figure 7). The percentage weight gain in non-treated and dextran treated mice was 9.5% and 4.41%, respectively. Our results are supported by a previous study in which EPS from Koper grains significantly reduced high-fat-diet-induced body weight gain in treated mice, compared to control, after three weeks [65]. Adult weight gain with body fat accumulation is a key determinant in the development of metabolic syndrome [66]. Moreover, rapid gestational weight gain is an early risk factor for later childhood obesity [67]. Obesity is a main risk factor of metabolic syndrome in children; in fact, further weight gain in obese children significantly reduces insulin sensitivity and worsens all components of metabolic syndrome [68]. Dietary modification is one preventative strategy to mitigate cardiovascular risk factors associated with metabolic syndrome [69]. Our findings suggest that supplementation with HBW1-derived dextran may prevent diet-induced weight gain. To the best of our knowledge, there are no earlier studies that utilize microbial dextran to investigate their role in weight gain.

#### 2.7.2. Investigation of Blood Glucose Level in Mice Treated with Bacterial Exopolysaccharide

The blood glucose level of the control group increased by 15.2% yet decreased by 16.26% in the dextran-fed group at the end of experiment (Figure 8). These results show that dextran have the capability to control the blood glucose level in mice, inferring their use as hypoglycemic agents. Our results are in accordance with those reported previously by Cao et al. [70], in which they used β-glucan in obese/type 2 diabetic mice and observed significant reduction in blood glucose levels. In another independent study, using different molecular weight polysaccharides from *Pseudostellaria heterophylla* were found highly effective in causing significant anti-hyperglycemic response in rats [71]. Poor glycemic control is highly associated with metabolic syndrome and acts as a predictor to cardiovascular disease [72]. Thus, the use of the investigated EPS as an antihyperglycemic agent would be beneficial for the management of metabolic syndrome. There are no data available on microbial dextran-type EPS regarding blood glucose studies. Therefore, the present study is the first to investigate the role of microbial dextran on blood glucose level.

#### 2.7.3. Investigation of Serum Cholesterol in Mice

The serum cholesterol level in HBW1 dextran-fed group was found lower than that of the control group. Dextran exhibited prominent hypocholesterolaemic effects as they reduced serum cholesterol level by 17.2%, in comparison to the control group (Figure 9). These results indicate the cholesterol-lowering potential of HBW1 dextran. Reduction in serum cholesterol by pharmaceutical agents is recommended as treatment for metabolic syndrome [9], suggesting the biotherapeutic potential of the novel dextran-type EPS. To the best of our knowledge, the effect of an α-glucan (e.g., dextran) on blood cholesterol level has not been studied previously. However, the cholesterol-reducing ability of an oat-based product fermented with β-glucan from *Pediococcus parvulus* 2.6 has been reported, which reduced total cholesterol levels in humans when compared to a control group [73]. Another study also reported the hypocholesterolaemic effect of β-cyclodextrin and the investigators found that daily consumption of 25 g/kg of 𝛽-cyclodextrin significantly reduced serum cholesterol by 25.9% in comparison with the control group [74]. Nevertheless, this is the first study in which α-dextran from a microbial source has been used to determine hypocholesterolemia activity in animal models.

## 3. Material and Methods

### 3.1. Screening and Identification of the Bacterial Isolate

The dextran type exopolysaccharide (EPS) synthesizing isolate *Apilactobacillus waqarii* (HBWI) used in the present study was isolated from honeybee gut (*Apis dorsata*) previously and reported as a novel species [75].

### 3.2. Bacterial Optimization of Culturing Parameters

For obtaining maximum yield of the EPS, different environmental parameters including aeration conditions, pH, temperature and substrate concentration were studied by the method previously used [76].

### 3.3. Extraction and Purification of EPS

Based on optimized conditions, the isolate HBW1 was incubated in MRS medium including 250 g sucrose in a shaking incubator (150 rpm) for 72 h at 30 °C. The EPS was purified according to previous methods [77].

### 3.4. Characterizations of EPS

#### 3.4.1. Determination of Sugar Composition and Moisture

The sugar composition was analyzed by the method previously described by Foyle et al. [78]. Briefly, EPS was hydrolyzed in 4% sulfuric acid and 0.1 g samples were loaded along with sugar recovery standard (glucose) in duplicates. The samples were autoclaved at 121 °C for 90 min and neutralized by adding barium carbonate (BaCO_3_) and then filtered. Sugar concentrations were quantified by a high-performance liquid chromatography (HPLC) system equipped with Aminex HPX-87H column which was pre-equilibrated with a mobile phase (0.005 M H_2_SO_4_). Analysis was performed at 50 °C; the injection volume was 5 μL with the flow rate of 0.6 mL/min. The data were collected through refractive index (RI) detector from which glucose recovery rate and glucan content were calculated using the following formulas.
$$\begin{array}{cc} \%Recovery & =\frac{Conc.\;detected\;by\;HPLC,\frac{\mathrm{mg}}{\mathrm{mL}}}{Conc.before\;hydrolysis,\frac{\mathrm{mg}}{\mathrm{mL}}}*100\%Recovery\\ & =\frac{Conc.\;detected\;by\;HPLC,\frac{\mathrm{mg}}{\mathrm{mL}}}{Conc.before\;hydrolysis,\frac{\mathrm{mg}}{\mathrm{mL}}}*100 \end{array}$$
where: Cx = Conc.sample, concentration of glucose in the hydrolyzed sample after correction for loss on acid hydrolysis, mg/mL.
$$\%Glucan=\frac{Cx*V\left(hydrolized\;liquid,\mathrm{mL}\right)*0.9}{Loading\;Weight*\left(100-moisture\%\right)/100}*100$$
where: 0.9 is the conversion factor for glucose to glucan.

Moisture level was determined in triplicates using standard method known as convection oven at (105 ± 3 °C for 4 h) by National Renewable Energy Laboratory (NREL), as used previously [79]. The following formula was used to calculate moisture content.
$$\%moisture=\frac{W\left(wet\;sample+dish,g\right)-W\left(dry\;sample+dish,g\right)}{W\left(wet\;sample+dish,g\right)-W\left(dish,g\right)}\;\times \;100\%$$

#### 3.4.2. ^13^C NMR Spectroscopy

^13^C NMR spectroscopy was performed to identify the glycosidic linkages in the *Apilactobacillus waqarii* EPS by the method previously used for microbial EPS [80].

#### 3.4.3. Methylation Analysis

The connectivity of glucose units in EPS from isolate HBW1 was confirmed by using the linkage analysis protocol previously performed [80]. Briefly, the purified EPS was permethylated with iodomethane and subsequently subjected to reductive cleavage by using sodium borodeuteride. Acetylation of the cleaved monomer units was performed by acetic anhydride. Data analyses were performed using gas chromatography–mass spectrometry (GC–MS).

#### 3.4.4. Molecular Weight Determination

The molecular weight of the EPS was determined using high performance size exclusion chromatography (HPSEC) with an RI detector, as previously described [81]. The data were collected and RI signals were interpreted.

### 3.5. Cell Line Experiments

Human colon (HT-29) cells were cultured in McCoy 5A media (Thermo Fisher Scientific, Waltham, MA, USA) supplemented with 10% FBS (Thermo Fisher Scientific, Waltham, MA, USA). Cells were used between passages 21 and 30 and grown at 37 °C in a 5% CO_2_ atmosphere. Media was changed every two to three days until cells reached confluency.

#### 3.5.1. Cell Viability and Cytotoxicity

The effect of the EPS on intestinal cell viability was investigated. Briefly, HT-29 cells were subcultured into a 96-well plate at a density of 20,000 cells/well and allowed to attach for 48 h. Four concentrations of EPS were prepared, including 0.5, 1, 3, and 5 mg/mL. McCoy 5A supplemented with 10% FBS was used as the control. Next, 100 μL of each treatment was added to the respective wells and incubated for 24 h at 37 °C. Viability of HT-29 cells was determined using methyl thiazolyl tetrazolium (MTT) solution from Bio Basic (Markham, ON, Canada). Manufacturer’s instructions were followed and absorbance read at 570 nm using a microplate reader. The results of the MTT assay are expressed as the mean percentage of cell viability relative to control, with error bars indicating the standard percent error among samples of each treatment. A one-way ANOVA was performed using Tukey’s post hoc analysis to account for between-sample comparisons (*n* = 5) and determine statistical significance (*p* < 0.05).

#### 3.5.2. Intracellular Reactive Oxygen Species (ROS) Assay

In order to assess the antioxidant potential of the novel EPS, a reactive oxygen species (ROS) assay was performed using intestinal epithelial cells. HT-29 cells (20,000 cells/well) were maintained in a 96-well plate until confluency was reached. In order to produce ROS, the cells were pre-treated with 100 ng/mL of interferon gamma (IFN-γ) for 12 h, followed by lipopolysaccharides (LPS; 100 ng/mL) for 24 h. LPS was then co-incubated with the EPS (2 mg/mL) for 24 h at 37 °C. McCoy 5A media supplemented with 10% FBS was used as the control. Relative expression of ROS following treatment was measured using an intracellular ROS assay kit, as instructed by the manufacturer (Abcam, Cambridge, UK). Deep red fluorescence was measured for individual samples (*n* = 5) and the mean relative ROS expression was plotted. A one-way ANOVA followed by Tukey’s post hoc analysis was used for comparison of means. Significance was measured at *p* < 0.05.

### 3.6. Experiments on Mice Models

#### 3.6.1. Formulation of Mice Feed

Dextran from *Apilactobacillus waqarii* HBW1 was synthesized by the method described above. The commercially available experimental mice feed was purchased and 3% (*w*/*w*) purified dextran was mixed for the experimental group. For the control group, the feed was used as purchased, without adding any other ingredient.

#### 3.6.2. Animals and Diet Design

Twelve laboratory five-week-old mice (BALB/c) were obtained from the National Institute for Biotechnology and Genetic Engineering (NIBGE), Faisalabad animal house. The mice were divided equally into two groups: control and dextran treatment groups. All the mice were given the control diet for five days prior to starting the experiment in order to help them acclimatize. After five days, the control group was continued to be fed with the same basic diet, while the experimental group received the basic diet additionally supplemented with 3% (*w*/*w*) dextran from *Apilactobacillus waqarii* HBW1. The mice were kept in their respective cages and the temperature was maintained as 27 ± 1 °C. The mice were given free access to feed and drinking water for 21 days.

#### 3.6.3. Determination of Body Weight

Body weight of both the groups were determined at baseline (Day 0) and at the end of the experiment (Day 21). For this purpose, each individual mouse was placed on a weighing scale and its weight was recorded in g.

#### 3.6.4. Blood Glucose Analysis

Glucose level was analyzed by using On Call glucometer (ACON Laboratories, Germany) at baseline (Day 0) and on the final day of the experiment (Day 21). The values were measured in milligrams per deciliter (mg/dL).

#### 3.6.5. Serum Cholesterol Analysis

At day 21, all the mice were anesthetized by Avertin (2,2,2-Tribromoethanol, Sigma–Aldrich, Yong-in, Gyeong-gi, Korea) and euthanized according to guidelines for handling laboratory animals [82]. Following euthanization, blood was collected in Gel and Clot activator tubes (Xinle, China) and centrifuged instantly at 4000 rpm for 15 min at 4 °C. The serum was collected and stored at −20 °C. Serum cholesterol was analyzed by Micro Lab-300 chemistry analyzer (Merck, Darmstadt, Germany) according to the protocol provided by the manufacturer. The values were recorded in mg/dL.

### 3.7. Statistical Analysis

Data are presented as mean ± standard deviation (SD). A one-way ANOVA was performed using Tukey’s post hoc analysis to account for between-sample comparisons and determine statistical significance (*p* < 0.05).

## 4. Conclusions

The increasing global burden of metabolic syndrome necessitates the discovery of novel biotherapeutics. The influential role of the gut microbiome in metabolic homeostasis presents a unique opportunity to target metabolic syndrome using microbiome-engineered therapeutics. The present study demonstrates the use of a newly isolated bacterial exopolysaccharide for the treatment of metabolic syndrome. This treatment successfully reduced body weight gain, serum cholesterol and blood glucose levels in animal models, while also improving viability and anti-oxidant activity in intestinal epithelial cells. Further research is needed to determine the dextran’s effect on gut microbiota modulation and advance understanding of its therapeutic mechanism of action for translational potential.

## Figures and Tables

**Figure 1 ijms-23-12692-f001:**
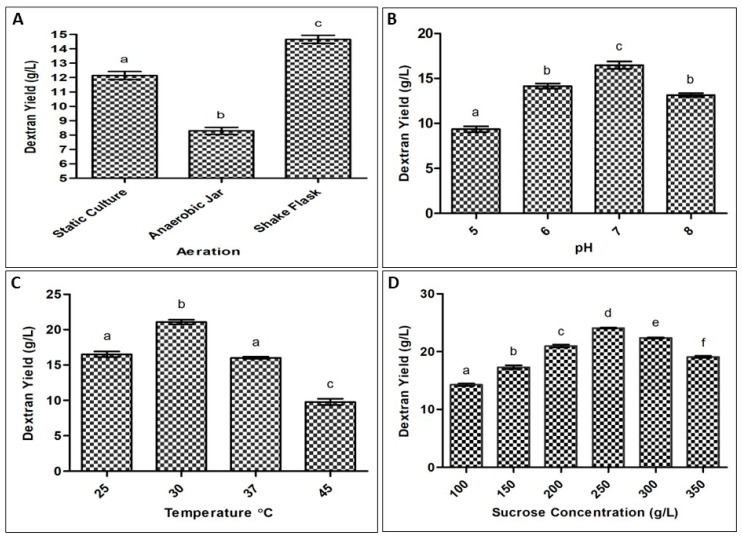
Effect of (**A**) aeration, (**B**) pH, (**C**) temperature and (**D**) substrate concentration on dextran production by *Apilactobacillus waqarii* after 72 h of incubation. The values of data sets (*n* = 2) denoted by different alphabet are significantly different (*p* < 0.05).

**Figure 2 ijms-23-12692-f002:**
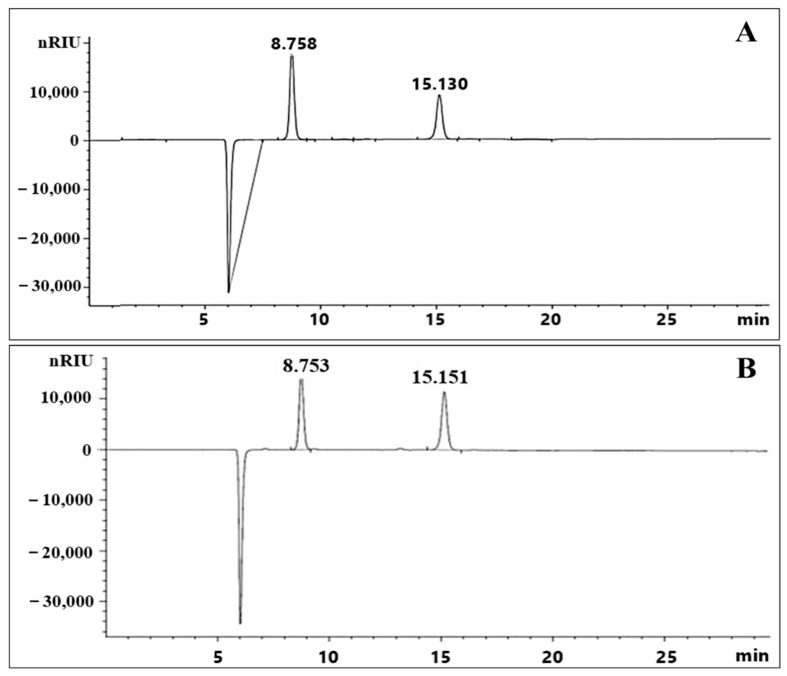
HPLC chromatogram of hydrolysate of dextran from (**A**) Standard and (**B**) Dextran from *Apilactobacillus waqarii*.

**Figure 3 ijms-23-12692-f003:**
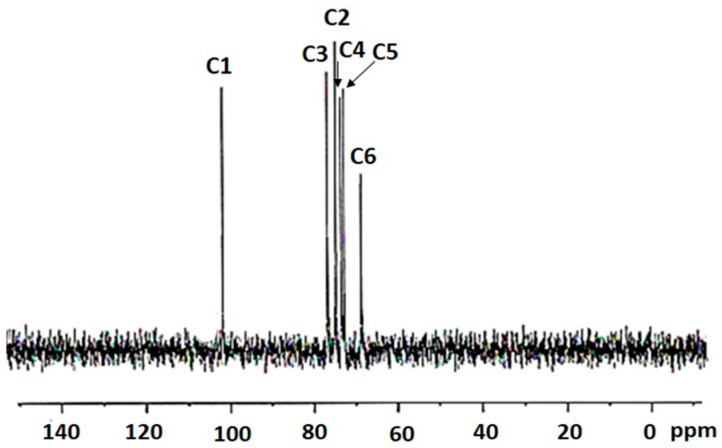
The ^13^C NMR spectra showing intensity peaks of the extracted EPS from *Apilactobacillus waqarii* in D_2_O.

**Figure 4 ijms-23-12692-f004:**
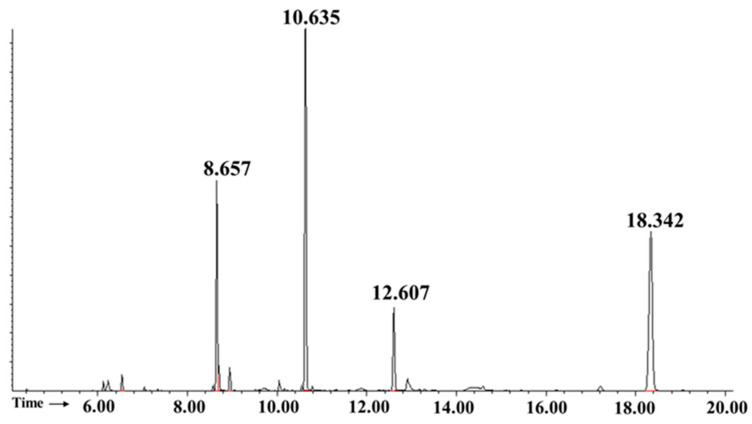
GC chromatogram from methylation analysis for *Apilactobacillus waqarii* dextran.

**Figure 5 ijms-23-12692-f005:**
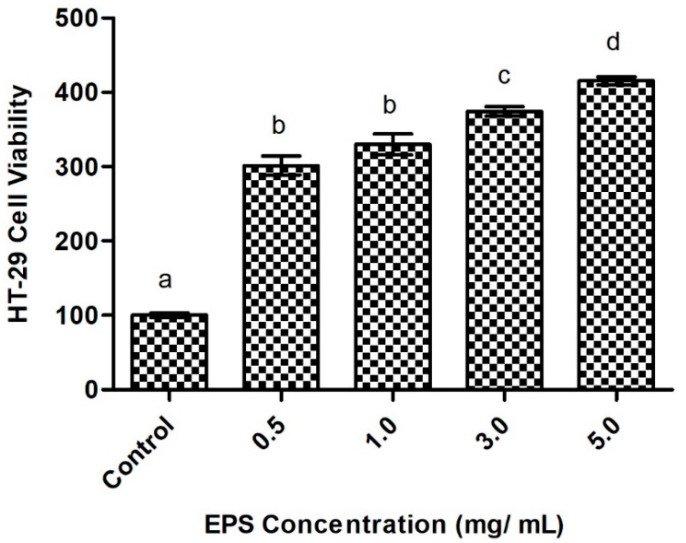
The effect of *Apilactobacillus waqarii* EPS on HT-29 intestinal cell viability at various concentrations. The values of data sets (*n* = 5) denoted by different alphabet are significantly different (*p* < 0.05).

**Figure 6 ijms-23-12692-f006:**
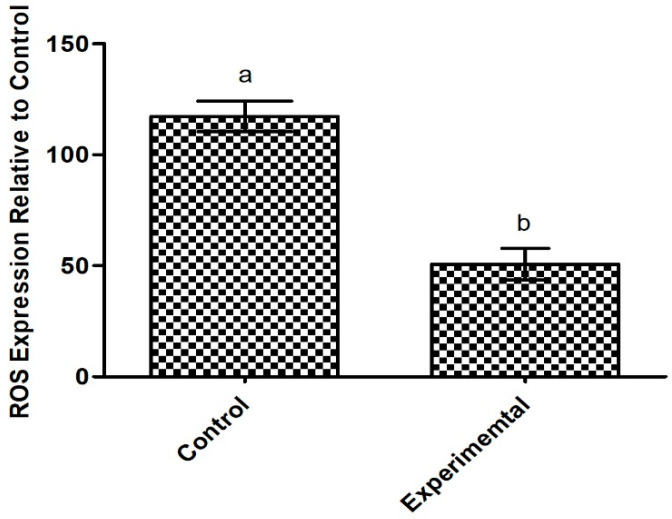
The expression of reactive oxygen species in HT-29 cells treated with *Apilactobacillus waqarii* EPS and control. The values of data sets (*n* = 5) denoted by different alphabet are significantly different (*p* < 0.05).

**Figure 7 ijms-23-12692-f007:**
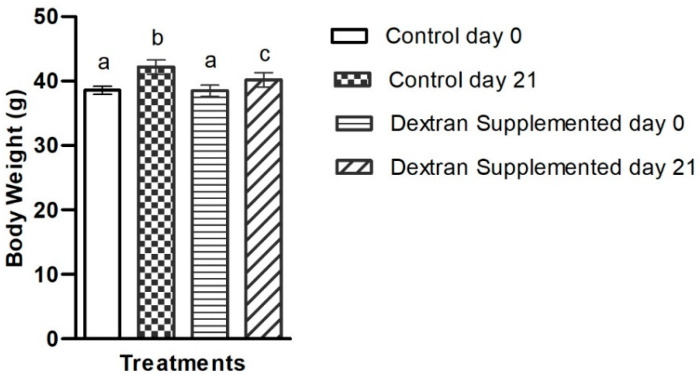
Average body weight gain in control and *Apilactobacillus waqarii* EPS supplemented group in comparison of day 0 and 21. The values of data sets (*n* = 6) denoted by different alphabet are significantly different (*p* < 0.05).

**Figure 8 ijms-23-12692-f008:**
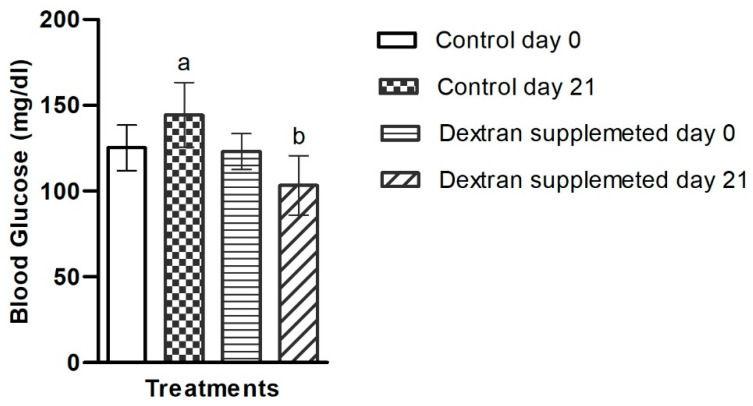
Blood glucose level of *Apilactobacillus waqarii* EPS in comparison with control at day 0 and 21. The values of data sets (*n* = 6) denoted by different alphabet are significantly different (*p* < 0.05).

**Figure 9 ijms-23-12692-f009:**
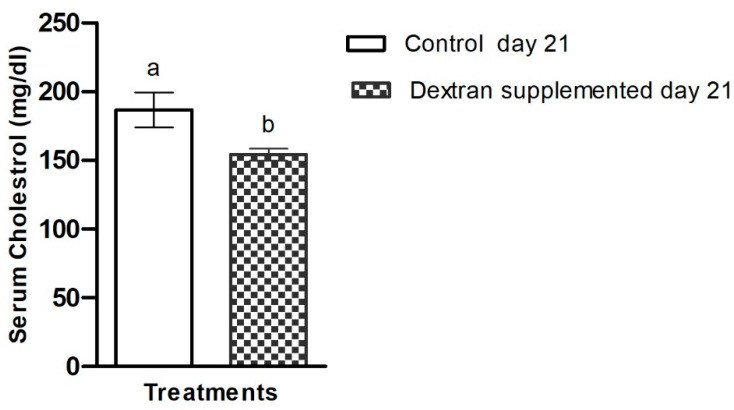
Serum cholesterol level of *Apilactobacillus waqarii* EPS with control in experimental animals. The values of data sets (*n* = 6) denoted by different alphabet are significantly different (*p* < 0.05).

**Table 1 ijms-23-12692-t001:** Linkage analysis of *Apilactobacillus waqarii* EPS by methylation and GC-MS.

Retention Time	Methylated Sugar	Primary MS Fragments (*m/z*)	Linkage Type	Molar Ratio (%)
8.657	1,5-Di-O-acetyl-2,3,4,6-tetra-O-methyl-D-glucitol	87, 102, 118, 129, 145, 161/162, 205	Terminal Glcp	21.5 ± 0.5
10.635	1,5,6-Tri-O-acetyl-2,3,4-tri-O-methyl-D-glucitol	87, 99, 102, 118, 129, 162, 189, 233	1→6-linked Glcp	66.74 ± 0.2
12.607	1,3,5,6-Tetra-O-acetyl-2,4-di-O-methyl-D-glucitol	87, 101, 118, 139, 160, 174, 189, 234, 305	1→3,6-linked Glcp	11.76 ± 0.2

## Data Availability

All the data supporting the findings of this study are available from the corresponding authors upon reasonable request.

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
