# Peer review of "High-Molecular-Weight Dextran-Type Exopolysaccharide Produced by the Novel *Apilactobacillus waqarii* Improves Metabolic Syndrome: In Vitro and In Vivo Analyses"

_ijms, 2022, doi:10.3390/ijms232012692_

Round 1

Reviewer 1 Report

It is an exciting and very original work; however, how it is presented can be improved.

1.- The introduction must be concise; it is too long.

2.- What is labeled as figures are graphs, and these graphs must be uniform with the same format throughout the writing, with their respective titles indicating how many experiments per group or the size of the groups in experimentation as well as the type of statistical test that was carried out, in addition to indicating if the standard error or the standard deviation is shown, and the p-value.

3.- It would be enriching to discuss the implications of EPS increasing viability in a dose-dependent manner in HT-29 cells because these cells derive from grade II colon adenocarcinoma. Similarly, the decrease in ROS.

4.- It could improve the understanding and importance of the work if the results are presented first and discussed until the end.

5.- Would it be appropriate to indicate what dietary supplements they contain?

6.- What was the exact diet of the control and experimental groups?

7.- It is not clear if the control group received a dietary supplement plus dextran or if they had a different diet.

7.- Why were insulin resistance and glycosylated hemoglobin not measured, and why was the body mass index not determined to reach a convincing conclusion?

8.- That obesity is a risk factor for metabolic syndrome is not necessarily a parameter to measure it.

Author Response

Please see attached file. Thank you. Best regards. Dr Prakash

Reviewer 2 Report

In general,

The manuscript entitled "High Molecular Weight Dextran-Type Exopolysaccharide Produced by The Novel Apilactobacillus waqarii Improves Metabolic Syndrome: In Vitro and In Vivo Analyses" is an interesting characterization of the newly isolated HBW1 for therapeutic benefits in metabolic syndrome acting as a promising biotherapeutic for the improvement of metabolic syndrome and its individual components.

The Instrument is well described, the Materials and Methods are clear and the Conclusions are valuable and useful for many in the field interested in the analysis.

However, it would be valuable to address the following comments:

The full name of HBW1 should be in italics? This is because in part of the writing it appears in italics and in others it does not.

In line 165 it should say Figure 3.

In line 238, in vivo and in vitro must be written in italics, homogenize this in all the writing where applicable.

Finally, it would be valuable to assign the significances with respect to the figures where applicable, that is, in the description of the figures, in the figures and in the figure captions, this when there is significance, this is important because it will help to point out the findings from the results not only in the discussion.

I remain at your service, greetings.

Author Response

Please see attached file. Thank you, Best regards. Dr Prakash

Round 2

Reviewer 1 Report

 The first part of the work is excellent; however, in the second part, they support the results with unclear statistical analysis.

They must specify in each group of comparisons if they did repetitions because it is unclear.

What normality tests were used to decide the use of parametric tests (ANOVA one-way)

The standard deviations are minimal; it seems that your experiments are done under ideal conditions, and this makes me doubtful.

So I would like to know if you can provide me with the raw data to verify your results, or please would it be appropriate to have it reviewed by an expert statistician?

Author Response

Thank you. 

Round 3

Reviewer 1 Report

The authors responded to comments appropriately.